# Effects of Enzymatically Induced Carbonate Precipitation on Capillary Pressure–Saturation Relations

Johannes Hommel [1,*], Luca Gehring [1], Felix Weinhardt [1], Matthias Ruf [2], Holger Steeb [2]

1 Department of Hydromechanics and Modelling of Hydrosystems, University of Stuttgart, Pfaffenwaldring 61, 70569 Stuttgart, Germany
2 Institute of Applied Mechanics, University of Stuttgart, Pfaffenwaldring 7, 70569 Stuttgart, Germany
* Correspondence: johannes.hommel@iws.uni-stuttgart.de

**Abstract:** Leakage mitigation methods are an important part of reservoir engineering and subsurface fluid storage, in particular. In the context of multi-phase systems of subsurface storage, e.g., subsurface $CO_2$ storage, a reduction in the intrinsic permeability is not the only parameter to influence the potential flow or leakage; multi-phase flow parameters, such as relative permeability and capillary pressure, are key parameters that are likely to be influenced by pore-space reduction due to leakage mitigation methods, such as induced precipitation. In this study, we investigate the effects of enzymatically induced carbonate precipitation on capillary pressure–saturation relations as the first step in accounting for the effects of induced precipitation on multi-phase flow parameters. This is, to our knowledge, the first exploration of the effect of enzymatically induced carbonate precipitation on capillary pressure–saturation relations thus far. First, pore-scale resolved microfluidic experiments in 2D glass cells and 3D sintered glass-bead columns were conducted, and the change in the pore geometry was observed by light microscopy and micro X-ray computed tomography, respectively. Second, the effects of the geometric change on the capillary pressure–saturation curves were evaluated by numerical drainage experiments using pore-network modeling on the pore networks extracted from the observed geometries. Finally, parameters of both the Brooks–Corey and Van Genuchten relations were fitted to the capillary pressure–saturation curves determined by pore-network modeling and compared with the reduction in porosity as an average measure of the pore geometry's change due to induced precipitation. The capillary pressures increased with increasing precipitation and reduced porosity. For the 2D setups, the change in the parameters of the capillary pressure–saturation relation was parameterized. However, for more realistic initial geometries of the 3D samples, while the general patterns of increasing capillary pressure may be observed, such a parameterization was not possible using only porosity or porosity reduction, likely due to the much higher variability in the pore-scale distribution of the precipitates between the experiments. Likely, additional parameters other than porosity will need to be considered to accurately describe the effects of induced carbonate precipitation on the capillary pressure–saturation relation of porous media.

**Keywords:** enzymatically induced carbonate precipitation; capillary pressure–saturation relation; pore-network modeling; computed tomography; microfluidics

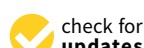



## 1. Introduction

Subsurface fluid storage is an important means to combat climate change by sequestering supercritical $CO_2$ [1] or storing energy, e.g., as compressed air, $CH_4$, or $H_2$, for coping with intermittent production of renewable sources, such as wind and solar, as well as balancing fluctuating supply and demand [2]. However, fluids stored underground have the potential to leak through damaged cap rocks or wellbores, posing a threat to other subsurface operations and the environment as well as reducing storage efficiency [3].

Engineered precipitation of minerals has the potential to seal such leakage pathways. Many minerals may precipitate due to a wide variety of biogeochemical reactions, which

can be influenced by engineering measures. The most widely used are carbonate minerals [4]. In this study, we will focus on the geo-engineered precipitation of induced calcium carbonate precipitation (ICP), which, among many other applications, has been demonstrated to have immense potential not only for subsurface engineering, but also for novel construction materials, soil stabilization, or remediation applications [4–8].

ICP reduces the porosity, permeability, and, likely, two-phase flow parameters of a porous medium by the precipitation of calcium carbonate, creating subsurface barriers for flow. Such barriers have the potential to remediate damaged cap rocks or leaky well cements in order to increase storage security by blocking potential leakage pathways. Most applications of ICP rely on urea hydrolysis by microbes or plant-based urease extracts to promote carbonate precipitation within porous media [9]. Urease is an abundant and well-studied enzyme [10], and urea is a relatively cheap reactant, which might explain the popularity of ureolytically induced carbonate precipitation. However, other metabolisms may also lead to the precipitation of carbonates [5]. In the presence of sufficient concentrations of calcium, this results in the overall reaction:

$$CO(NH_2)_2 + 2\,H_2O + Ca^{2+} \xrightarrow{\text{urease}} 2\,NH_4^+ + CaCO_3 \downarrow;\qquad(1)$$

Several large- or field-scale demonstrations of ICP have been completed with the aims of modifying soil properties, e.g., [11–16], or mitigating leakage in deeper subsurface, e.g., [8,17–21]. Often, these demonstrations were assisted by numerical investigations, e.g., [14,22,23]. Additionally, numerical investigations are used to study the feasibility of leakage mitigation in the deep subsurface, such as [24,25]. While ICP is widely proposed as a leakage mitigation technology, only a few field-scale investigations were actually conducted in the relevant two-phase flow conditions. The first step in the experimental investigation of two-phase flow was the lab-scale investigation of microbial ICP in [26]. The field application of Kirkland et al. [27] goes one step further, as it successfully applied ICP to seal leakage pathways in the presence of a $CO_2$ phase at the field scale.

However, most models developed of ICP assume full saturation of the porous medium with an aqueous phase [28–33]. Most recently developed models, especially those explicitly designed with leakage mitigation in mind, consider the impact of ICP on the porous medium's hydraulic properties, but this is usually limited to predicting the permeability change based on the change in porosity. Even those models of ICP that consider two-phase flow, e.g., [34–36], generally do not account for ICP-induced changes of two-phase flow parameters, such as the capillary pressure–saturation relation. A simplistic approach is used in [34] to account for ICP changing capillary pressure–saturation relations using Leverett scaling [37]. However, even in those instances where an impact of ICP on hydraulic parameters is considered in numerical models, the parameterizations are usually rather simplistic. Permeability changes are mostly accounted for using a power law, e.g., [34–36], or relations based on the Kozeny–Carman equation, e.g., [32,33].

The effect of ICP on porosity–permeability relations has become the target of experimental studies with the aim of understanding the effect of ICP or similar precipitation processes at the pore scale [38,39], expanding on previous microfluidic studies of ICP, e.g., [40–43]. Both [38] and [39] investigated porosity–permeability relations, observing the porosity change by microscopy and calculating the permeability change from pressure measurements. However, these quasi-2D geometries allow for convenient and time-resolved imaging but usually cannot represent realistic 3D geometries as they occur in real porous media [44]. Recently, porosity–permeability relations, not only of ICP, have been studied in 3D systems. The use of micro X-ray computed tomography (µXRCT) to observe the porosity change has been conducted, e.g., [45,46]. µXRCT eliminates the geometrical constraints of 2D microfluidic systems, providing more realistic porosity–permeability relations for use in predictive numerical models. As a recent example, ref. [47] examines the flow properties of various sandstones by using pore-network modeling on pore networks extracted from µXRCT images. Experimentally, the impact of pore-geometry changes has been investi-

gated in, e.g., carbonate dissolution in [48]. Similarly, ref. [49] investigated the effect of $CO_2$ injection into carbonate rocks on multi-phase flow properties.

The respective two- or multi-phase flow relations, such as capillary pressure–saturation or relative-permeability-saturation relations, have not yet been studied in detail, even though they are arguably of equal importance for the understanding and numerical modeling of leakage mitigation or other reservoir applications. Capillary pressure $p_c$ is generally defined as the difference between the non-wetting phase pressure $p_n$ and the wetting phase pressure $p_w$:

$$p_c = p_n - p_w. \tag{2}$$

On the scale of a single pore throat or capillary tube with radius $r_c$, the individual capillary pressure can be calculated based on the Young–Laplace equation:

$$p_c = \frac{2\gamma \cos \Theta}{r_c}, \tag{3}$$

where $\gamma$ is the interfacial tension between the wetting and non-wetting fluids, and $\Theta$ is the contact angle between the fluid–fluid interface and the solid surface. Based on the bundle of capillary tubes model, which simplifies a porous medium into a bundle of parallel tubes, e.g., [50], and the Young–Laplace equation, which shows the effect of radius on capillary pressure, it can be expected that any precipitation within porous media that necessarily, at least locally, reduces pore or pore-throat radii, and thereby also the capillary pressure in this pore or pore throat, is likely to influence capillary pressure–saturation relations on the representative elementary volume (REV) scale. For porous media in general, however, consideration of individual pores becomes impractical, necessitating the use of alternative descriptions, e.g., algebraic relations, or based on assumptions of bundles of capillary tubes through which the wetting phase saturation for a given capillary pressure is determined by integration, as, e.g., completed for dynamic conditions by [51]. The result of this approach is then a relation between the wetting phase saturation and the capillary pressure, which represents Equation (3) on a larger scale using averaging over a REV. Such REV-scale capillary pressure–saturation relations are widely used in numerical modeling of two- or multi-phase flow in porous media. The commonly used relations for modeling are of the Brooks–Corey [52] and Van Genuchten types [53]. The relation according to Brooks–Corey [52] is:

$$p_c = p_e S_e^{-\frac{1}{\lambda}}; \text{ for } p_c > p_e, \tag{4}$$

and the relation according to Van Genuchten is:

$$p_c = \frac{1}{\alpha}\left(S_e^{-\frac{1}{m}} - 1\right)^{\frac{1}{n}}; \text{ for } p_c > 0, \tag{5}$$

both provide an equation to estimate $p_c$ based on the effective wetting phase saturation $S_e$, which can be calculated based on the current and residual water saturations $S_w$ and $S_{w,r}$, respectively:

$$S_e = \frac{S_w - S_{w,r}}{1 - S_{w,r}}. \tag{6}$$

For the Brooks–Corey relation, the additional parameters are the entry pressure $p_e$ and the parameter $\lambda$. For the Van Genuchten relation, the parameters are $\alpha$, $m$, and $n$. The Van Genuchten parameter $m$ is sometimes expressed as $m = 1 - 1/n$ for theoretical considerations, enabling complete integration of Mualem's equation [54].

We use two different experimental setups as the basis for this: the 2D microfluidic experiments of [38], which provide excellent possibilities for observation by optical microscopy with a resulting high temporal resolution but at limited realisticity of the porous medium, and the 3D sintered glass-bead columns, which offer more realistic porous media; however, this is more difficult to image as it requires the much more time-consuming

lab-based µXRCT scanning as the imaging method, which does not allow for temporal resolution within experiments. Both setups were mineralized using enzymatically induced calcium carbonate precipitation (EICP), and, while the 2D microfluidic cells were continuously imaged, the 3D columns could only be imaged after mineralization. The columns with an initial average porosity ranging from 36% to 39% were thus mineralized to various degrees to achieve different porosity reductions. The decision to stop the mineralization treatment was based on the increase in differential pressure to 1 bar, 8 bar, and 8 bar at a reduced flow rate, resulting in a 21, 56, and 59% porosity reduction for the low, medium, and high mineralizations. The mineralized 3D columns were then scanned in a custom µXRCT device with a single micrometer resolution, as described in detail in [55]. The reconstructed images of the 3D columns were finally segmented into three phases: glass, void, and precipitate. See the published datasets [56–59]. The continuously imaged 2D microfluidic cells were selected at various steps of porosity reduction and segmented into solid and pore space; the actual pore geometry was reconstructed based on previous investigations [38]. See also the dataset [60].

To capture and later upscale what are essentially pore-scale displacement processes, it is important to investigate the scale of the pores. Direct pore-scale modeling using the segmented images directly, e.g., by [61,62], or, in general, methods not requiring morphological assumptions, e.g., [63,64], result in the most realistic results, capturing the effects of sub-pore-scale surface roughness or additional sub-pore-scale complexity, such as, e.g., the heterogeneous distribution of contact angles [65]. However, direct pore-scale simulations on the actual geometry as determined by the XRCT images of the 3D samples, or even the comparatively small geometry of the 2D samples, are likely prohibitive in computational cost, especially when using the entire domain or large parts of it. Thus, there is a need to simplify the imaged geometry without sacrificing too much of the detailed geometrical knowledge of the image data. This can be achieved by extracting a pore network (PN) from the available segmented images. While a PN represents a simplified geometry that, obviously, cannot represent every sub-pore-scale detail, by choosing angular pore-body and pore-throat geometry, which better approximate the angular pore geometry of our samples, wetting fluid flow is possible even after the non-wetting fluid has invaded a location due to wetting fluid remaining connected via the corners [50]. This will account for some, but not all, sub-pore-scale detail, but we prefer investigating a larger domain over having the highest resolution, as our aim is ultimately to upscale to the REV-scale, a transition in which some detail of even the pore scale might be lost. Pore-network modeling (PNM) combines a relatively realistic, though simplified, version of the pore-scale geometry of a porous medium, idealized into a network of pore bodies linked by pore throats [50,66]. PNs represent a scale between the exact pore scale and coarser scales, and PNM can be a convenient tool for determining coarser-scale, upscaled properties, such as, e.g., intrinsic permeability, but also multi-phase flow properties [46,50,67–69]. PNM has been used to investigate the impact of simultaneous mineral dissolution and precipitation on porosity–permeability relations for certain hypothetical scenarios of pore-geometry change in, e.g., [70]. PNM has also been used to investigate reactive transport affecting pore geometry by, e.g., mineral dissolution and precipitation [71] or biofilm growth [72,73]. Moreover, in the context of ICP, PNM has been used, e.g., to determine the interplay of pore-scale transport and reactions in [74].

We extracted PNs from the segmented images of both the complete 2D and 13 lengthwise sections of each of the 3D samples, using the complete diameter at the given porosity reductions. This way, we ensured that our simulation domains are large enough to be considered a representative elementary volume (REV) while retaining a geometry that is as realistic as possible. We prefer using PNs extracted from the actual geometry over PNs created on a structured lattice based on a statistical analysis of the actual geometries, as proposed by, e.g., [75]. On the extracted PN, we evaluate capillary pressure–saturation curves by performing numerical primary drainage experiments using water and air as the wetting and non-wetting phase to determine the effect of EICP on the capillary pressure–

saturation relation. Both the Brooks–Corey [52] and Van Genuchten [53] types of capillary pressure–saturation relations are fitted to the curves obtained by the numerical primary drainage experiments.

Finally, we study the impact of the porosity reduction due to precipitation on the parameters of both the Brooks–Corey ($p_e$, $\lambda$, $S_{w,r}$) and the Van Genuchten ($\alpha$, $m$, $n$, $S_{w,r}$) capillary pressure–saturation relations to find suitable parameterizations for implementation in numerical models describing ICP. In general, we expect to observe an increase in capillary pressure for a given saturation and a decrease in porosity during mineralization with ICP as the precipitates reduce the space available for the fluids and reduce the pore-throat radii for which Equation (3) predicts higher capillary pressures at given saturations. Additionally, Leverett scaling [37] suggests increased capillary pressure for a given saturation, when the porosity and, therefore, also the permeability are reduced. For constant fluid and fluid–solid-interaction properties, Leverett scaling results in scaling the capillary pressure for the reference condition $p_{c,0}$ by a factor determined by the changes in porosity $\phi$ and permeability $K$:

$$\frac{p_c(\Delta\phi)}{p_{c,0}} = \sqrt{\frac{K_0\phi}{K\phi_0}}, \tag{7}$$

where $p_c(\Delta\phi)$ is the capillary pressure for the porosity change $\Delta\phi$, and $K_0$ and $\phi_0$ are the initial values of permeability and porosity. A problem with using Leverett scaling is that it also requires a description of the effect of ICP on the permeability $K$, which is not a trivial task, as observed by several experimental studies, e.g., [38,39,41].

In this study, we attempt the first exploration of the effect of ICP on capillary pressure–saturation relations, combining experimental observations in microfluidic systems as a base and PNM as a method to evaluate the effect of the observed pore-geometry changes due to ICP on the capillary pressure–saturation curves. Investigating different degrees of porosity reduction allows us to propose capillary pressure–saturation–porosity relations that, on an REV scale, describe the effects of ICP on capillary pressure. This enables us to describe such effects on the REV-scale models of ICP, which are commonly used to design, monitor, and assess the real-world applications of ICP, thereby improving the predictive capabilities of such models for leakage mitigation scenarios, which are a common scenario for the use of ICP. We believe this study is a crucial step toward more realistic models of ICP in the context of subsurface leakage mitigation applications and an improved general understanding of the impact of ICP on two- or multi-phase flow.

## 2. Materials and Methods

### 2.1. Samples

We used two types of samples: quasi-2D, microfluidic glass cells, further called 2D for their convenience in imaging, and 3D sintered glass-bead columns, further called 3D for their geometry that better represents bulk porous media.

As the quasi-2D structure of glass microfluidic cells is more accessible for imaging, such microfluidic glass cells were used to enable a high temporal resolution. The chemically wet-etched glass cells were purchased from Micronit©, Enschede, The Netherlands, and are made out of Borosilicate glass. The glass cells feature two inlet and outlet channels, distribution channels, and a pore structure in the center of the glass cells. Each of the inlet and outlet channels, respectively, connects to a pressure sensor in order to measure the differential pressure across the pore structure. The solid matrix of the pore structure consists of pillars of various sizes with diameters ranging from 200 to 700 µm. Initially, thus, the pore space is fully connected and of a constant height of 35 µm. The porous domain in this case has the dimensions 20.5 mm × 11.9 mm × 0.035 mm. More details regarding the microfluidic cells can be found in [38].

The 3D columns consist of sintered borosilicate glass beads with a mean diameter of 180 µm. The overall dimensions of the columns include a diameter of $d = 5$ mm and a length of $l = 10$ mm. The columns were sealed on the side with shrink tubes and, using

epoxy resin, glued into the center of a cylindrical, 3D-printed plastic mold. This was done to ensure the consistent outer dimensions of each sample and to create a wide enough surface around the inlet and outlet ends of the columns to prevent leakage or flows that bypass the o-ring seals placed in the sample holder. Figure 1 shows the steps of the sample preparation and the sample holder.

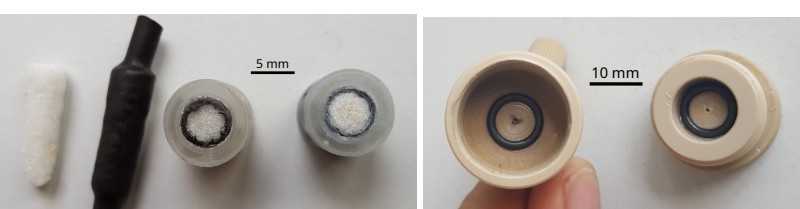

**Figure 1.** 3D column sample preparation from left to right: sintered glass-bead column, column wrapped in shrink tube, and columns fixed in the plastic mold; both clean and mineralized (**left**) and the sample holder (**right**).

**Preparation of reactive solutions:**

Two reactive solutions were prepared as described in [39]: a reactant solution containing urea and calcium as well as a urease solution. The reactant solution was prepared by dissolving urea and calcium chloride (both MERCK©, Darmstadt, Germany) in deionized water at equimolar concentrations of $1/3$ mol/L. The urease solution was prepared by suspending jack-bean meal (Sigma Aldrich©, Darmstadt, Germany) in deionized water at a concentration of 5 g/L, which was stirred at 8 °C for 17 h and then filtered through a 0.45 µm cellulose membrane to eliminate remaining solids.

*2.2. Experimental Setup And Procedure*

The setup for the 2D samples is sketched in Figure 2. The quasi-2D microfluidic cells were initially saturated with deionized water. The reactant and urease solutions were co-injected into the glass cells using DC motor-driven syringe pumps (mid-pressure pumps, type neMESYS 100N) from CETONI GmbH, Korbussen, Germany, at controlled flow rates using 2.5 mL glass syringes. Both solutions mix in a T-junction before entering the microfluidic cell through the inlet. The outlet was connected to a constant-head reservoir elevated 10 cm above the cell to create a back pressure to reduce gas bubble formation. A pressure sensor (maximum pressure of 70 mbar, resolution of 16 bit, accuracy of 14 Pa, and internal volume of 7 µL) from ELVEFLOW, Paris, France, was connected to each of the second inlet and outlet connections of the microfluidic cell to measure the differential pressure across the porous structure. The precipitation process was observed continuously by optical microscopy. In parallel to imaging, the flow and pressure data were logged continuously (see [38]). This imaging approach ensured a high temporal and spatial resolution with the drawback of only providing 2D images. The initial flow rate was 40 nL/s, and this was decreased to 20 nL/s once the limit of the pressure sensor had been reached. This was repeated once more, and, finally, the mineralized glass cell was flushed with deionized water to stop further reaction. During the 2D experiments, the ambient temperature was 25 °C. Note that Experiment 2D1 required restarting, including resaturation, after approximately 26 h due to clogging issues in the inlet zone of the cell. The relatively high flow rates during the resaturation process may have led to experimental artifacts, such as the initiation of a preferential flow path. More details on the procedure of the 2D experiments can be found in [38]. The arrangement of the 2D setup is shown in Figure 3.

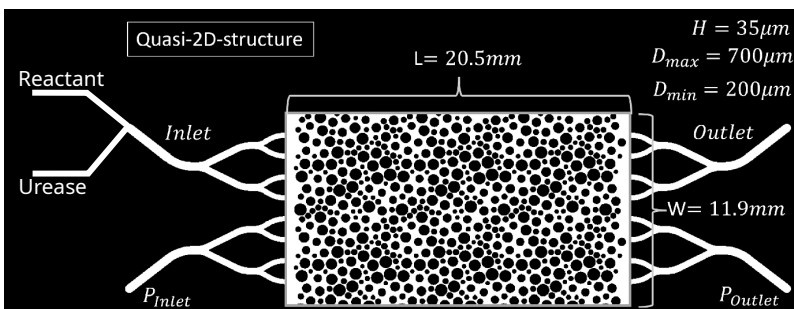

**Figure 2.** Sketch of the 2D EICP setup as used in this study, modified from [38].

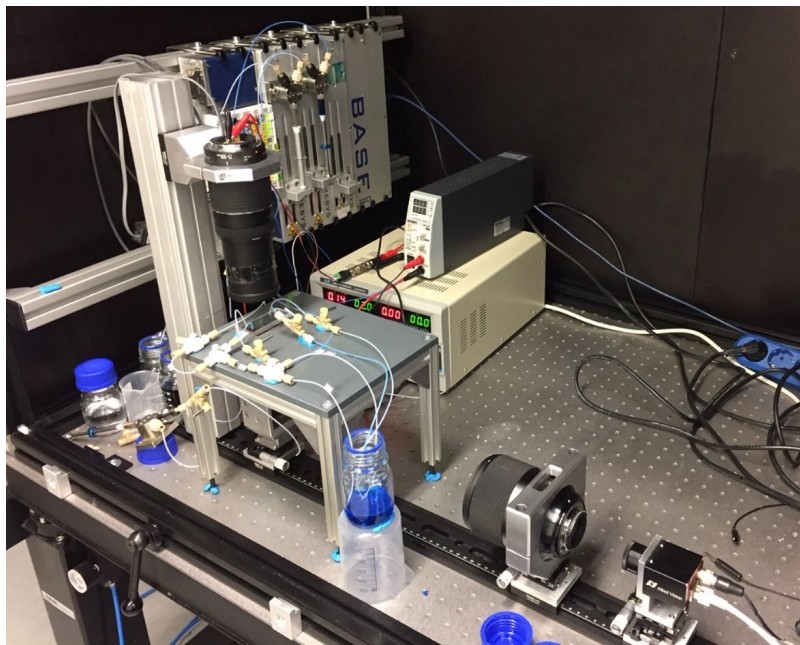

**Figure 3.** Photograph of the 2D setup as used during the EICP-mineralization experiments.

The setup for the 3D samples is, in general, similar to the 2D samples. Each glass-bead column was placed in a sample holder with three inlets. Two inlets were connected to the two syringe pumps, as in the 2D setup, but, here, we use larger, 5 ml glass syringes. As in the 2D experiments, one syringe was filled with the reactant solution containing calcium chloride and urea, and the other one was filled with the urease solution containing urease. The third inlet was connected to a pressure sensor (maximum pressure of 8 bar, resolution of 12 bit, accuracy of 2 mbar) from CETONI GmbH, Korbussen, Germany. The outlet was connected to a waste container with a constant head, and the diameter of the outlet tube was large enough (1.6 mm) to neglect the pressure drop along its length, which was calculated to be less than 10 Pa. Therefore, the pressure measured at the inlet subtracted by the constant head pressure of the outlet can be assumed to be the pressure difference over the column at a given flow rate. However, the pressure was mainly measured to monitor the progress of mineralization in the glass-bead column samples, rather than to accurately determine the permeability of the pore structure. The setup with the installed 3D samples was initially saturated with 70% ethanol and then flushed with deionized water to disinfect the sample and tubing as well as to avoid initial gas bubbles.

The mineralization of the glass-bead column was promoted by co-injecting the urease as well as the urea and calcium chloride solutions into the glass-bead columns. Both solutions were injected at a constant rate of 5 µL/s each, resulting in a total injection rate of 10 µL/s. During the entire mineralization process, the inlet pressure was monitored. Further, the glass-bead columns were immersed during mineralization in a water bath at 60 °C to increase the reaction rates and achieve sufficient amounts of precipitation within a

working day. In the experimental study [76], 20 g/L was hydrolyzed almost completely within 120 min at 60 °C, while, at 30 °C, only about half of that was hydrolyzed within the same time by identical urease concentrations. Three columns were mineralized to various degrees with the aim of achieving varying degrees of porosity reduction: The first was mineralized to a differential pressure of approximately 1 bar. The second was mineralized to a differential pressure of approximately 8 bar, and the third was mineralized to a differential pressure of approximately 8 bar, but with a step-wise reduced injection rate. Each time the 8 bar pressure threshold was reached, the injection rate was reduced to the final injection rate of 0.5 μL/s.

After the mineralization process, the system was flushed with deionized water in order to replace the reactive solutions inside the column and avoid further precipitation. Imaging of the 3D samples was carried out after mineralization.

### 2.3. Image Acquisition

The images of the 2D samples were acquired using optical microscopy. The microscopy setup used was as described by [77], with slight modifications for the needs of the EICP setup. The frame rate was between 0.1 and 1 fps, but only images at select 2D porosity reduction steps were chosen for further processing. The image resolutions were 2296 × 1349 pixel and 2289 × 1348 pixel (length × width) for the two 2D experiments 2D1 and 2D2, respectively.

To characterize the inner 3D structure of the three prepared glass-bead column samples after the completed mineralization procedure, micro X-ray computed tomography (μXRCT) imaging was performed. For this, the modular and open micro X-ray computed tomography (μXRCT) system, described in [55], was used. Except for the geometric magnification, all three samples were scanned with the same settings: X-ray tube voltage 90 kV, X-ray tube flux 90 kV, detector exposure time 2000 ms (Shad-o-Box 6K HS detector employed), and 1800 equidistant projection angles with 5 slightly in-plane shifted detector positions for bad detector pixel compensation, cf. [55]. In the μXRCT datasets analyzed in this contribution, the medium-mineralization sample was imaged with a geometrical magnification of $M = 26.18$, and the high- and low-mineralization samples with $M = 22.51$, which leads to corresponding uniform voxel edge lengths of 1.9 μm and 2.2 μm in the reconstructed datasets. The physical size of the field of view is about 5.59 mm × 5.59 mm × 4.07 mm and 6.47 mm × 6.47 × 4.71 mm (width × depth × height). Consequently, the complete cross-section area (diameter 5 mm) of the prepared samples may be visualized; however, the entire height of the sample cannot. Since the field of view in the vertical direction is not large enough, it was focused on the inlet for samples 3 and 10 and on the outlet for sample 4. The 3D volume reconstruction was performed with the software Octopus Reconstruction (Version 8.9.4-64 bit) [78] using the Feldkamp–Davis–Kress (FDK) algorithm [79] for the cone-beam reconstruction. Typical artifacts, such as beam hardening and ring artifacts, were reduced with the implemented methods. The underlying datasets (projection, reconstructed images), including further metadata, can be found in [56–58].

### 2.4. Image Post-Processing

After obtaining the images, post-processing was necessary to evaluate the change in pore geometry and the resulting effects on the upscaled parameters' porosity and capillary pressure–saturation relation.

#### 2.4.1. 2D Experiments

As the 2D experiments were imaged in a high temporal resolution and the initial, empty porous geometry was available, it was sufficient to segment the images into void and solid, as the precipitated $CaCO_3$ can be distinguished by subtracting the initial, empty porous geometry as a mask. Further, as the obtained images were only in 2D, the actual 3D geometry had to be reconstructed from the 2D images for a more realistic account of the geometry change.

Segmentation

The gray-scale images obtained from optical microscopy were segmented to determine the geometry change due to EICP. For this, selected images were processed with the software Matlab R2019b$^{©}$ (The Mathworks, Inc., Natick, MA, USA) [80]. Before the actual image processing, the initial, unmineralized porous domain was set as a mask and all subsequent images were registered to the initial image. The main processing consisted of smoothing, morphological, and, finally, binarization operations. When comparing images at different time steps while precipitation is still taking place, it is important that they are geometrically aligned. In the resulting binarized images, individual crystal aggregates can be identified, and the size of their 2D projection can be derived. More details on image processing are given in [38,39].

3D Reconstruction from 2D Images

The 3D information was reconstructed from the 2D images obtained by optical microscopy, as described in more detail in [38]. Based on an assumption of a characteristic, representative crystal shape, the area of a crystal aggregate on the 2D image and the height of the microfluidic cell are used to determine the 3D reconstruction of the precipitate within the microfluidic cell and thereby the pore geometry. Compared to a final µXRCT scan, frustum and spheroidal shapes were shown to be the best choices to determine the volume from a projected area, with the spheroidal shape only being applicablem while single, separated crystals could be identified [38]. Thus, we use a frustum shape with a slope of $\alpha_f = 72°$ for reconstruction in this study as it showed the best agreement in volume reconstruction in [38]. For further details regarding the reconstruction, we refer to [38].

2.4.2. 3D Experiments

As imaging of the 3D experiments was only carried out after the mineralization, our intent was to segment the µXRCT images not only into void and solid, but to additionally separate the solid into glass beads and precipitated $CaCO_3$. With this three-phase segmentation, at least the initial and the final pore geometry within the samples are available for analysis and do somewhat compensate for the lacking temporal resolution as in the easy-to-image 2D experiments.

Segmentation

As the 3D experiments were imaged after the mineralization was completed, the images had to be segmented into three phases: glass beads, $CaCO_3$, and void. Distinguishing between glass beads and $CaCO_3$ was possible due to a slight contrast in their gray values. However, at the interfaces of both phases, the noise within the gray values of the images prohibits a simple, threshold-based segmentation, which would lead to an unrealistic, ragged interface between areas segmented as glass beads and $CaCO_3$. Further, as the intensity of glass beads is lower than the intensity of $CaCO_3$, any simple, threshold-based segmentation would lead to the occurrence of a thin glass layer at the $CaCO_3$–void interface, which is not only highly unrealistic but also makes it impossible to use the segmented images to evaluate the unmineralized initial geometry by accounting only for the glass phase as a solid and $CaCO_3$ and void combined as the initial void space.

Due to these reasons, segmentation of the µXRCT slices was carried out using the machine learning-based software ilastik [81]. Ilastik was developed for image analysis, mainly in a microbiological context, and offers sophisticated pixel and object classification, amongst many other image analysis tools. Using ilastik, the abovementioned artifacts can be minimized by simple, threshold-based segmentation as the machine-learning classification by ilastik also takes into account the surrounding neighboring pixels. Ilastik was trained by indicating on a few slices which features belong to which phases by coloring an increasing number of features in the software until a satisfactory segmentation was achieved throughout the column. The output of ilastik are maps of the probability of segmentation into the three phases: glass beads, $CaCO_3$, and void, as shown in Figure 4.

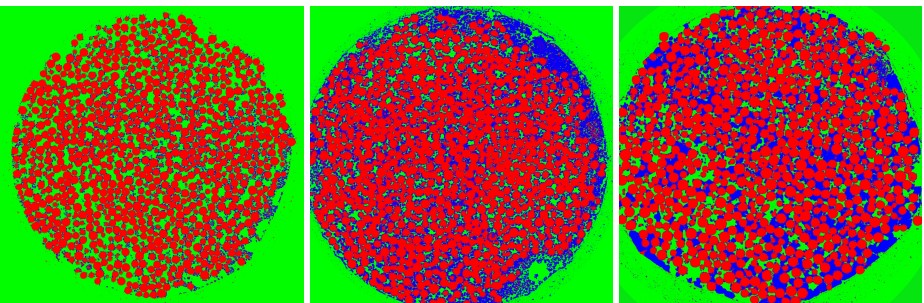

**Figure 4.** Distribution of the segmented phases, glass (in red), void (in green), and CaCO$_3$ (in blue), for the 3D experiments over selected cross sections of the three sample columns: slice #570 for the low mineralization (**left**), slice #477 for the medium mineralization (**center**), and slice #706 for the high mineralization (**right**).

The probability distributions from ilastik are further processed by smoothing operations using Matlab R2019b$^©$ (The Mathworks, Inc.) [80], similarly to the processing of the 2D experiments (see Section 2.4.1). Additionally, a circular mask was overlaid over the slices to eliminate areas outside of the column as well as the rough and irregular edges of the sample columns, limiting the investigation in this study to the regular center of the column to avoid influences by boundary effects.

### 2.5. Pore-Network Generation

Pore-network geometries were extracted from the segmented images using the PoreSpy python toolkit [82,83], which segments the images using a marker-based watershed algorithm. The parameters used for the 2D experiments were $R = 10$ and $\sigma = 20$, while, for the 3D experiments, $R = 5$ and $\sigma = 1$ were used, where $R$ is the radius of a structuring element of the watershed algorithm used in PoreSpy, and $\sigma$ is the standard deviation of the convolution kernel used for smoothing to eliminate spurious peaks.

### 2.6. Pore-Network Modeling

The PNs extracted from the processed images of both the 2D and 3D experiments were used to evaluate capillary pressure–saturation curves on the geometry affected by EICP by performing numerical primary drainage experiments to determine the effect of EICP on the capillary pressure–saturation curves. In these numerical primary drainage experiments, the pore network is initially saturated with water during the wetting phase. On one side, air is introduced during the invading non-wetting phase, and the global capillary pressure between water and air is increased step by step from 0 Pa to the maximum pressure, which was 10 kPa for the 2D experiments and 20 kPa for the 3D experiments. At each pressure step, the global capillary pressure is compared to the pore throat's entry pressure for all pore throats at the interface between air and water. Whenever the pore throat's entry pressure is exceeded, air will invade the pore throat, until all pore throats at the air–water interface have entry pressures higher than the globally applied capillary pressure. Then, we integrate the phase volumes over the pore network to determine the phase saturations. By repeating these steps and recording the pairs of applied capillary pressure and resulting saturations, we can construct the capillary pressure–saturation curves of each pore network, representing a simplified version of the geometry of each of the samples.

PNM was carried out using the release version 3.4 of the simulator DuMu$^\text{x}$ [84]. DuMu$^\text{x}$ (Dune for Multi-Phase, Component, Scale, Physics, ... flow and transport in porous media) is a free and open-source simulator [85]. We use the pore-network models as implemented and published previously by [86–88]. The specific code used with installation instructions was published on 30 May 2022 at: https://git.iws.uni-stuttgart.de/dumux-pub/gehring2022a. The results of the PNM with the geometry reconstruction and the extracted PN are published in the datasets [59,60].

## 3. Results

In the following, we present the results of our study, first for the 2D experiments and second for the 3D. For each set of experiments, we give the capillary pressure–saturation curves resulting from the numerical primary drainage experiments as well as the Brooks–Corey- and Van Genuchten-type capillary pressure–saturation relations fitted to parameterize them. For the 2D experiments with high temporal and, thus, porosity reduction resolution, we also parameterize the effect of the porosity reduction on the parameters of the obtained Brooks–Corey- and Van Genuchten-type relations.

### 3.1. 2D Experiments

Figure 5 shows the extracted pore network overlaid over the microscopy image of the unmineralized 2D sample. The capillary pressure–saturation curves obtained by PNM for the various porosity reductions as well as the Brooks–Corey and Van Genuchten relations fitted to them are shown in Figure 6 for the 2D1 sample and in Figure 7 for the 2D2 sample.

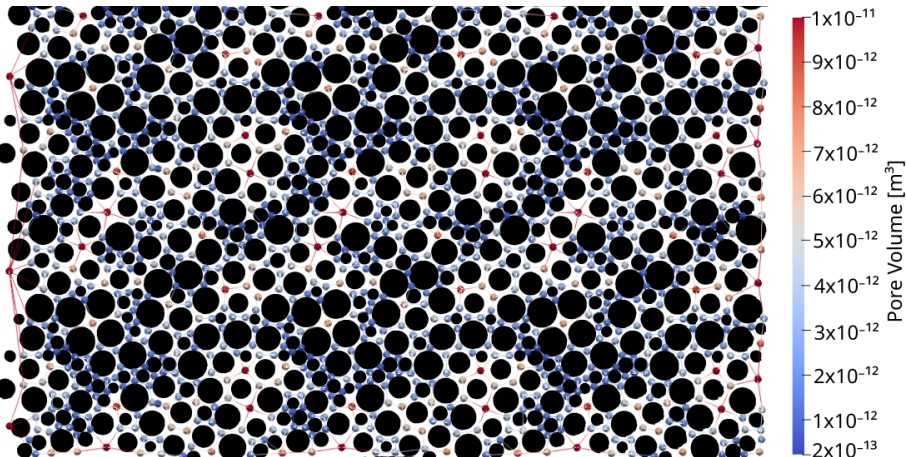

**Figure 5.** Extracted pore network showing the determined pore volumes overlaid over the microscopy image.

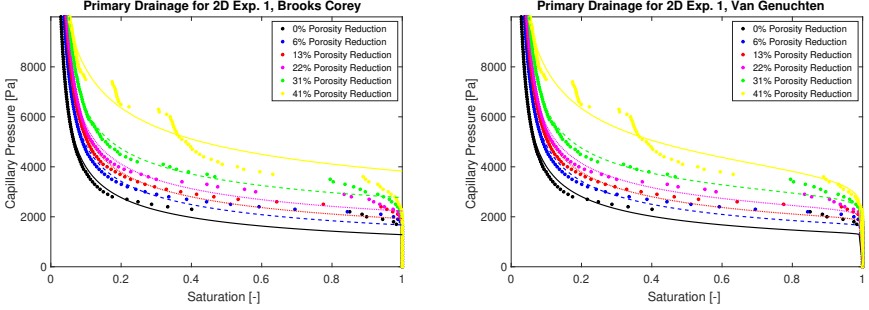

**Figure 6.** Capillary pressure–saturation relations for Experiment 2D1 at various porosity reductions as fitted with Brooks–Corey (Equation (4)) (**left**) and Van Genuchten relation (Equation (5)) (**right**).

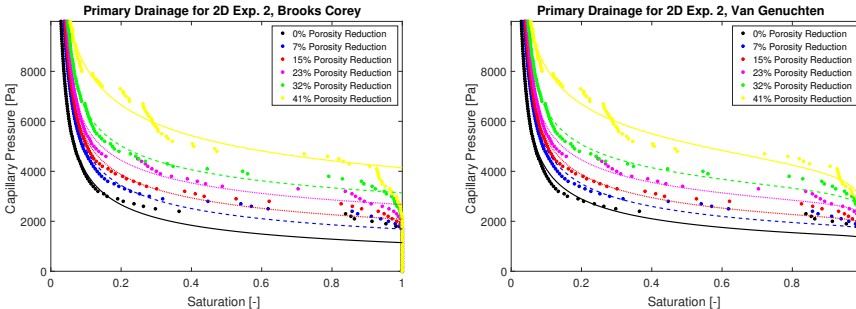

**Figure 7.** Capillary pressure–saturation relations for Experiment 2D2 at various porosity reductions as fitted with Brooks–Corey (Equation (4)) (**left**) and Van Genuchten relation (Equation (5)) (**right**).

As can be seen in Table 1 and Figures 8 and 9, both 2D experiments show similar behavior of the fitted Brooks–Corey-relation parameters; in general, all increase with increasing porosity reduction. Likely due to the homogeneous thickness of the microfluidic cells, at least initially, the residual saturation is relatively small, $S_r < 4\%$. The residual saturation increases only slightly with an increasing $\Delta\phi$, except for the highest porosity reduction of 41%, while $\lambda$ and $p_e$ increase significantly with $\Delta\phi$. However, the changes are, in general, smaller in Experiment 2D1, in which a preferential flow path remained largely unmineralized, compared to Experiment 2D2 with more homogeneous precipitation [38]. The Brooks–Corey-relation parameter $\lambda$ increases from 2.2 and 2.0 to 3.17 and 3.39 as the porosity reduction by ICP increases from 0% to 41%, for Experiments 2D1 and 2D2, respectively. Similarly, the entry pressure $p_e$ increases from 1284 Pa and 1145 Pa to 3829 Pa and 4143 Pa as the porosity reduction by ICP increases from 0% to 41%, for Experiments 2D1 and 2D2, respectively.

**Table 1.** Parameters of the Brooks–Corey capillary pressure–saturation relation (Equation (4)) for various porosity reductions $\Delta\phi$ due to EICP.

| $\Delta\phi$ [-] | $\lambda$ [-] | $p_e$ [Pa] | $S_r$ [-] |
|---|---|---|---|
| | | Experiment 2D1 | |
| 0.0 | 2.195 | 1283.5 | 0.0185 |
| 0.06 | 2.407 | 1673.1 | 0.0252 |
| 0.13 | 2.594 | 1929.7 | 0.0303 |
| 0.22 | 2.875 | 2230.6 | 0.0332 |
| 0.31 | 3.328 | 2765.9 | 0.0382 |
| 0.41 | 3.172 | 3829.2 | $1.57 \times 10^{-8}$ |
| | | Experiment 2D2 | |
| 0.0 | 1.986 | 1145.4 | 0.0160 |
| 0.07 | 2.425 | 1692.3 | 0.0249 |
| 0.15 | 2.815 | 2086.0 | 0.0295 |
| 0.23 | 3.484 | 2672.9 | 0.0358 |
| 0.32 | 3.792 | 3140.9 | 0.0372 |
| 0.41 | 3.391 | 4142.7 | $5.21 \times 10^{-9}$ |

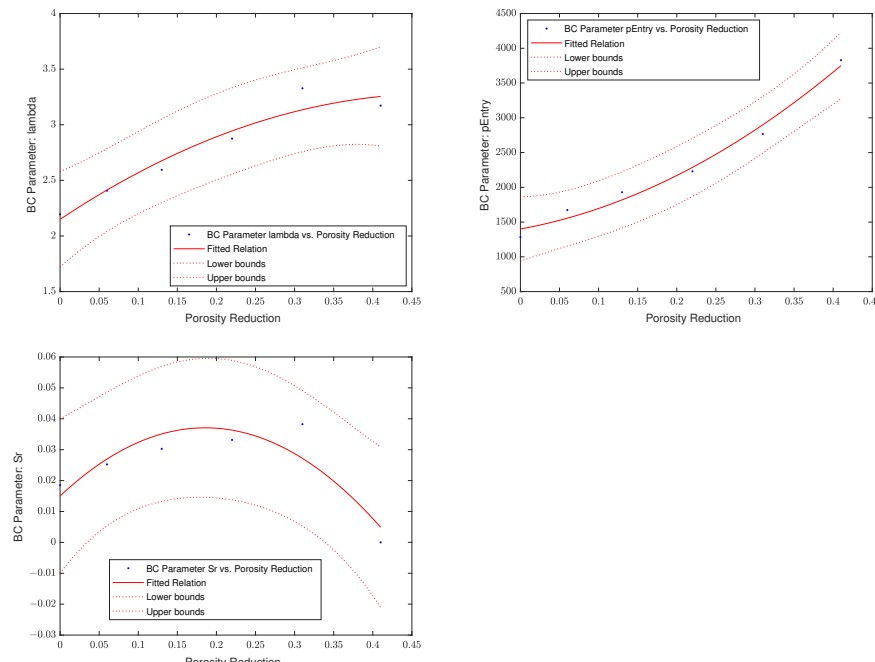

**Figure 8.** Observed dependency of the Brooks–Corey parameters on the porosity reduction $\lambda$ (**left**), entry pressure $p_\mathrm{e}$ (**right**), and residual saturation $S_\mathrm{r}$ (**bottom**) for Experiment 2D1.

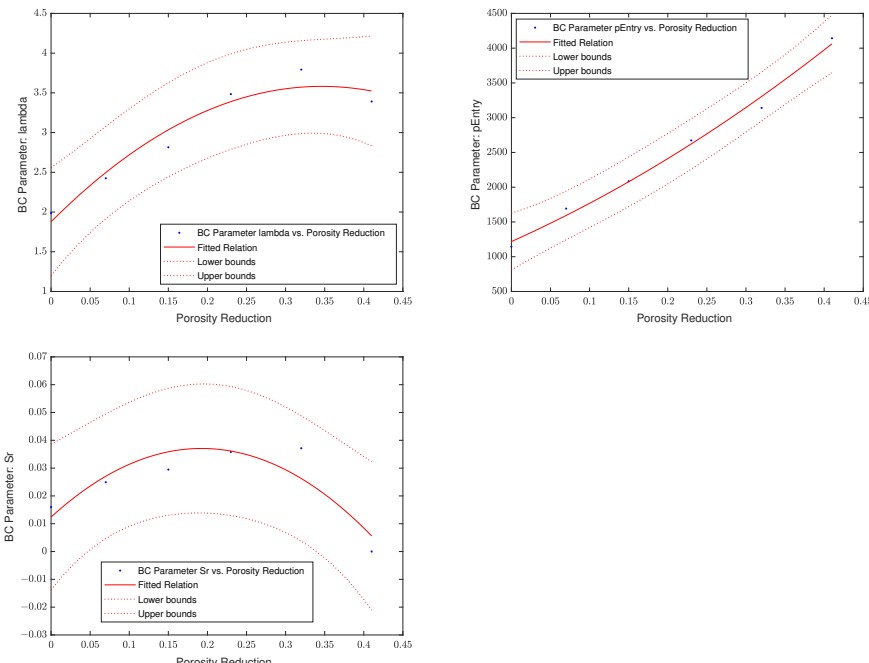

**Figure 9.** Observed dependency of the Brooks–Corey parameters on the porosity reduction $\lambda$ (**left**), entry pressure $p_\mathrm{e}$ (**right**), and residual saturation $S_\mathrm{r}$ (**bottom**) for Experiment 2D2.

As can be seen in Table 2 and Figures 10 and 11, both 2D experiments show similar behavior of the fitted Van Genuchten-relation parameters: $m$ increases with increasing porosity reduction, while $n$ and $\alpha$ decrease. Using Van Genuchten capillary pressure–saturation relations, the determined residual saturations are again very low and increase slightly with increasing porosity reduction, behaving very similarly to the fitted Brooks–Corey capillary pressure–saturation relations, see Table 1. Again, the highest porosity reduction of 41% presents a kind of exception with $S_\mathrm{r} \approx 0$ for Experiment 2D1, and

$S_{\rm r} = 0.01$ for Experiment 2D2, as this is lower than all other fitted residual saturations. Interestingly, the residual saturation for Experiment 2D2 at $\Delta\phi = 41\%$, $S_{\rm r} = 0.01$, is surprisingly different from the one determined for the same geometry when fitting a Brooks–Corey-type relation. Further, the residual saturations of Experiment 2D1 seem to be relatively similar, independent of the choice of the fitted capillary pressure–saturation, while the values for $S_{\rm r}$ vary more for Experiment 2D2 depending on the relation used (see Tables 1 and 2).

**Table 2.** Parameters of the Van Genuchten capillary pressure–saturation relation (Equation (5)) for various porosity reductions $\Delta\phi$ due to EICP.

| $\Delta\phi$ [-] | $\alpha$ [1/Pa] | $m$ [-] | $n$ [-] | $S_{\rm r}$ [-] |
|---|---|---|---|---|
| | | Experiment 2D1 | | |
| 0.0 | $7.53 \times 10^{-4}$ | 0.0397 | 56.87 | 0.0191 |
| 0.06 | $5.94 \times 10^{-4}$ | 0.0375 | 64.62 | 0.0254 |
| 0.13 | $5.17 \times 10^{-4}$ | 0.0632 | 41.15 | 0.0303 |
| 0.22 | $4.38 \times 10^{-4}$ | 0.1235 | 23.92 | 0.0338 |
| 0.31 | $3.56 \times 10^{-4}$ | 0.1910 | 17.80 | 0.0388 |
| 0.41 | $2.63 \times 10^{-4}$ | 0.3105 | 10.11 | $3.08 \times 10^{-8}$ |
| | | Experiment 2D2 | | |
| 0.0 | $7.13 \times 10^{-4}$ | 0.0409 | 57.09 | 0.0196 |
| 0.07 | $5.60 \times 10^{-4}$ | 0.0517 | 49.42 | 0.0262 |
| 0.15 | $4.77 \times 10^{-4}$ | 0.0831 | 34.05 | 0.0296 |
| 0.23 | $3.70 \times 10^{-4}$ | 0.1688 | 20.92 | 0.0361 |
| 0.32 | $3.08 \times 10^{-4}$ | 0.2665 | 15.00 | 0.0384 |
| 0.41 | $2.29 \times 10^{-4}$ | 0.3974 | 9.723 | 0.0107 |

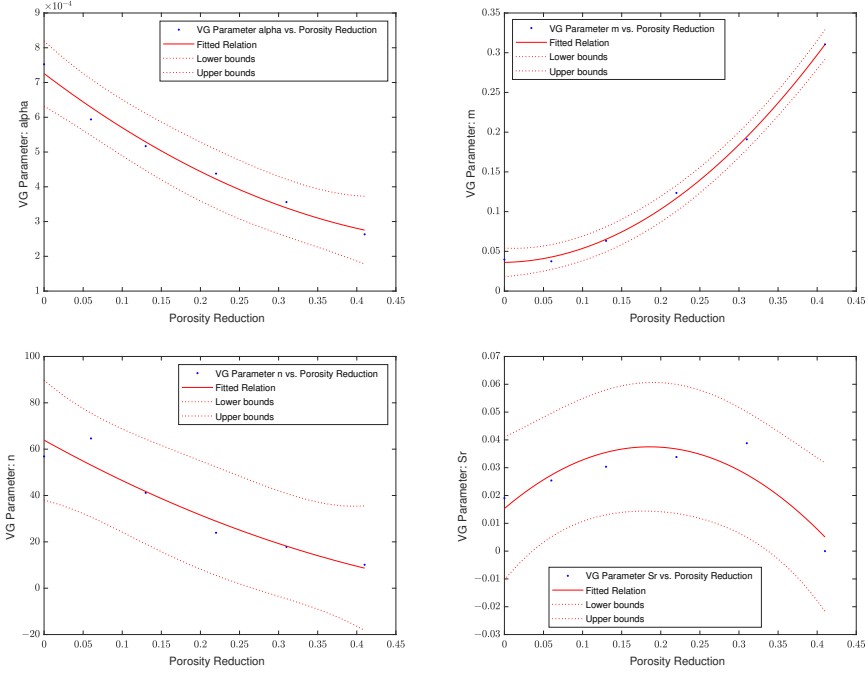

**Figure 10.** Observed dependency of the Van Genuchten parameters on the porosity reduction: from top left to bottom right, $\alpha$, $m$, $n$, and residual saturation $S_{\rm r}$ for Experiment 2D1.

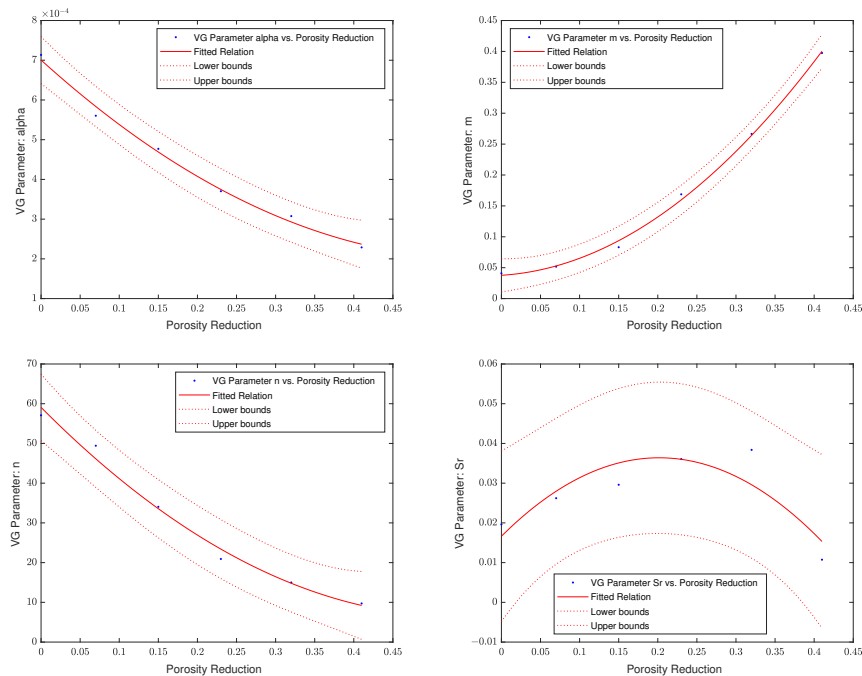

**Figure 11.** Observed dependency of the Van Genuchten parameters on the porosity reduction: from top left to bottom right, $\alpha$, $m$, $n$, and residual saturation $S_\mathrm{r}$ for experiment 2D2.

To better quantify the effect of the porosity reduction $\Delta\phi$ by EICP on the parameters of both the Van Genuchten and Brooks–Corey parameterizations for the capillary pressure–saturation relation for porous media, see Equations (4) and (5), in which the observed parameters $p$ at various porosity reductions $\Delta\phi$ are fitted using a second-order polynomial expression:

$$p = a \cdot \Delta\phi^2 + b \cdot \Delta\phi + c, \tag{8}$$

where $a$, $b$, and $c$ are the fitted coefficients of the polynomial expression, and the coefficients have the same unit as the parameter being fitted. The motivation for choosing a quadratic expression is that it is the simplest non-linear expression. The values for the coefficients are shown in Table 3.

**Table 3.** Parameters of the second-order polynomial expression describing the effect of porosity reduction due to EICP on the capillary pressure–saturation relations, see Equation (8).

| Parameter | $a$ | $b$ | $c$ |
|---|---|---|---|
| | Experiment 2D1 | | |
| $\lambda_\mathrm{2D1}$ | $-4.8389$ | $4.6779$ | $2.1494$ |
| $p_\mathrm{e,2D1}$ | $8963.3$ | $2056.8$ | $1400.4$ |
| $S_\mathrm{r,2D1}$ | $-0.6398$ | $0.2377$ | $0.0150$ |
| $\lambda_\mathrm{2D2}$ | $-14.228$ | $9.8493$ | $1.8770$ |
| $p_\mathrm{e,2D2}$ | $4619.1$ | $5043.8$ | $1218.0$ |
| $S_\mathrm{r,2D2}$ | $-0.6651$ | $0.2562$ | $0.0124$ |
| | Experiment 2D2 | | |
| $\alpha_\mathrm{2D1}$ | $1.48 \times 10^{-3}$ | $-1.71 \times 10^{-3}$ | $7.26 \times 10^{-4}$ |
| $m_\mathrm{2D1}$ | $1.5901$ | $0.0169$ | $0.0361$ |
| $n_\mathrm{2D1}$ | $127.61$ | $-187.05$ | $63.887$ |
| $S_\mathrm{r,2D1}$ | $-0.6443$ | $0.2392$ | $0.0153$ |
| $\alpha_\mathrm{2D2}$ | $1.59 \times 10^{-3}$ | $-1.78 \times 10^{-3}$ | $7.00 \times 10^{-4}$ |
| $m_\mathrm{2D2}$ | $1.9645$ | $0.0788$ | $0.0377$ |
| $n_\mathrm{2D2}$ | $186.22$ | $-197.95$ | $59.077$ |
| $S_\mathrm{r,2D2}$ | $-0.4855$ | $0.19585$ | $0.0166$ |

### 3.2. 3D Experiments

The 3D experiments have much larger numbers of pores and pore throats compared to the 2D experiments. Due to the significant numbers of pores and pore throats, the pore networks were evaluated for 13 sections of each column, consisting of 100 slices of the segmented µXRCT images each. Effects of mineralization similar to the 2D setups can be observed for the 3D setups, as shown in Figure 12. However, the results are more complex for the 3D setups. We observe differences not only between the sample columns themselves but also heterogeneity along the length of each of the columns.

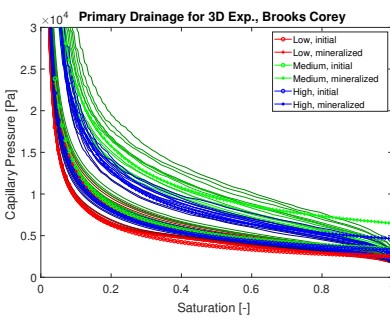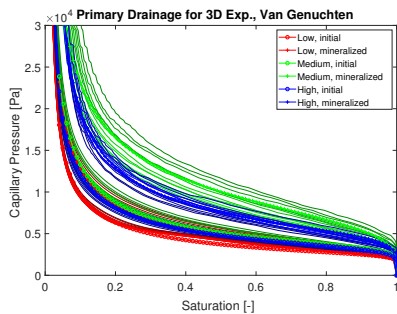

**Figure 12.** Capillary pressure–saturation curves for 13 sections of the mineralized and unmineralized 3D experiment columns each with Brooks–Corey (Equation (4)) (**left**) and Van Genuchten relations (Equation (5)) (**right**) fitted to each ensemble of capillary pressure–saturation curves obtained for each column in unmineralized and mineralized states.

The capillary pressure increases with mineralization. However, the changes in the capillary pressure–saturation relations are not as straightforward as for the initially identical 2D setups. While the unmineralized capillary pressure–saturation relations of both the high- and medium-mineralization columns are relatively similar, the unmineralized low-mineralization column's capillary pressure is significantly lower for the same saturation. As expected of the low amount of mineralization, the low-mineralization column's capillary pressure is the lowest of all mineralized columns; it is approximately equal to those of the unmineralized high- and medium-mineralization columns. However, after mineralization, the medium-mineralization column has by far the highest capillary pressure. During the mineralization of the 3D experiments, the differential pressure was the only indication of the progress of mineralization (see Section 2.2). The initial unmineralized porosities of the 3D samples are not completely identical but relatively similar $36.2\% < \phi_0 < 39.1\%$ (see Table 4). The difference between the final mineralized porosity and the porosity reduction in the high- and medium-mineralization 3D samples is quite small, even though we aimed at mineralizing the samples to various degrees. The volume fraction of $CaCO_3$ is identical for the high- and the medium-mineralization samples, but, due to the lower initial unmineralized porosity, the high-mineralization sample still has the highest porosity reduction at $\Delta\phi = 59.5\%$.

In addition to the porosities, Table 4 gives the mineralized permeability, normalized with the unmineralized permeability, which can be used together with the mineralized and unmineralized porosities of the samples to calculate the Leverett-scaling factor according to Equation (7). Due to the significant reduction in permeability of several orders of magnitude measured in the experiments, Leverett scaling according to Equation (7) predicts significant increases in capillary pressure by factors from 7.3 to 43.6 with mineralization due to EICP.

**Table 4.** Unmineralized ($\phi_0$) and mineralized porosities ($\phi$), average CaCO$_3$ volume fraction ($\phi_c$), porosity reduction ($\Delta\phi$), the normalized permeabilities of the mineralized sample ($K/K_0$), and the Leverett-scaling factor ($p_c/p_{c,0}$) computed according to Equation (7) of the 3D experiments.

| Sample | $\phi_0$ [-] | $\phi$ [-] | $\phi_c$ [-] | $\Delta\phi$ [-] | $K/K_0$ | $p_c/p_{c,0}$ |
|---|---|---|---|---|---|---|
| low | 0.391 | 0.309 | 0.082 | 0.211 | 0.015 | 7.3 |
| medium | 0.382 | 0.167 | 0.215 | 0.563 | 0.002 | 14.8 |
| high | 0.362 | 0.147 | 0.215 | 0.595 | 0.000214 | 43.6 |

One reason for the differences in capillary pressure–saturation relations, especially those of the unmineralized samples, might be that the matrix of porous glass beads was not perfectly homogeneous for all columns, leading to pore volume and, likely, connectivity variations along the length of the columns (see Figure 13). The volume fractions shown in Figure 13 were determined within a circular mask excluding the irregular edge of the sample and the area outside the actual sample to avoid artifacts. This variation complicates the parameterization and quantification of the impact of mineralization on the capillary pressure–saturation relations for the 3D samples and denies a simple parameterization of the impact of mineralization on the parameters of capillary pressure–saturation relations.

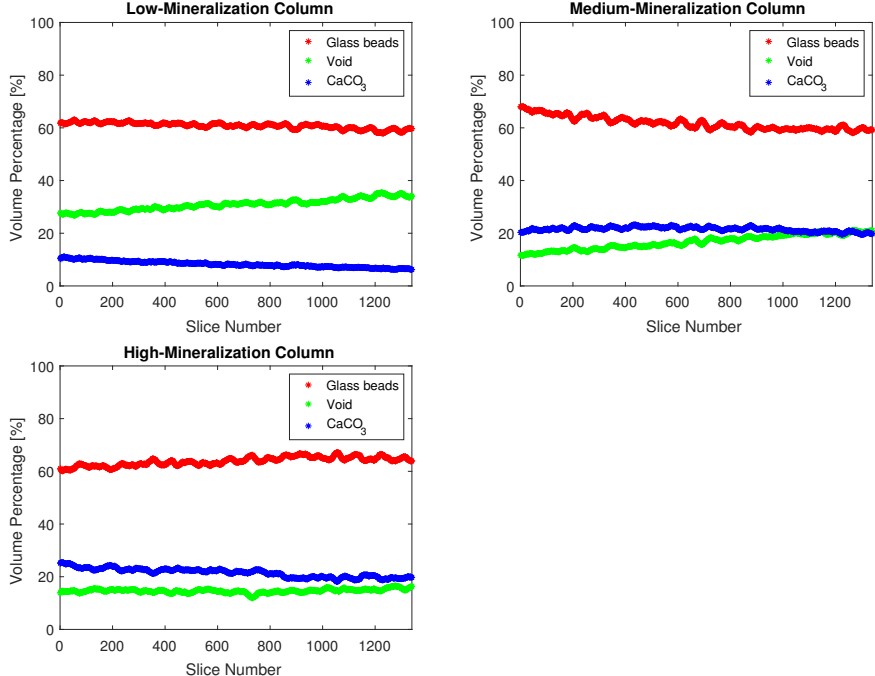

**Figure 13.** Volumes of the segmented phases (glass, void, and CaCO$_3$) for the 3D experiments and their distribution along the height of the three sample columns: low (**left**), medium (**right**), and high mineralization (**bottom**).

Upon close inspection of the segmented images, as shown in Figure 4, for selected cross sections of the columns, the precipitation pattern of the high- and medium-mineralization columns is different. For the medium-mineralization column, the precipitation is distributed in numerous, small, individual precipitates, while for the high-mineralization column most precipitates are much larger, though less numerous, and often completely fill the spaces between glass beads. Such distributions of precipitates result in larger, but fewer, remaining pores for the high-mineralization column and smaller, but more numerous, pores and pore throats for the medium-mineralization column, which could explain the observed differences in the capillary pressure–saturation curves and the fitted capillary pressure–saturation relations.

Due to the significant heterogeneity along the length of the columns, pore networks were evaluated for the 13 sections of each column, consisting of 100 slices of the segmented

µXRCT images each. Tables 5 and 6 give the parameters for the Brooks–Corey and the Van Genuchten relations as fitted to the determined capillary pressure–saturation curves of each of the 13 sections of the 3D samples as well as the parameters for the average relations fitted to the curves of all 13 sections of each column combined, as shown in Figure 12.

**Table 5.** Parameters of the Brooks–Corey capillary pressure–saturation relation (Equation (4)) for the initial and mineralized 3D experiments as fitted to each of the 13 sections along the column length.

| Column | Section | Initial | | | Mineralized | | |
|---|---|---|---|---|---|---|---|
| | | $\lambda$ [-] | $p_e$ [Pa] | $S_r$ [-] | $\lambda$ [-] | $p_e$ [Pa] | $S_r$ [-] |
| high | 1 | 1.9198 | 3071.1 | 0.0129 | 1.9093 | 4588.9 | 0.0288 |
| | 2 | 1.9416 | 3164.6 | 0.0137 | 1.8904 | 4443.3 | 0.0305 |
| | 3 | 1.9381 | 3244.5 | 0.0139 | 1.7862 | 4587.8 | 0.0228 |
| | 4 | 1.9584 | 3368.2 | 0.0142 | 1.7180 | 4708.6 | 0.0156 |
| | 5 | 1.9463 | 3284.1 | 0.0136 | 1.7357 | 4624.4 | 0.0194 |
| | 6 | 1.9492 | 3367.1 | 0.0134 | 1.6171 | 5057.2 | 0.0042 |
| | 7 | 1.9252 | 3616.9 | 0.0150 | 1.4828 | 4777.9 | $1.09 \times 10^{-8}$ |
| | 8 | 1.9681 | 3480.7 | 0.0137 | 1.6805 | 4932.8 | 0.0076 |
| | 9 | 1.9734 | 3591.2 | 0.0142 | 1.6274 | 4834.9 | 0.0132 |
| | 10 | 1.9675 | 3629.4 | 0.0142 | 1.7802 | 4755.2 | 0.0216 |
| | 11 | 1.9870 | 3669.7 | 0.0141 | 1.6393 | 4800.2 | 0.0053 |
| | 12 | 1.9683 | 3636.3 | 0.0143 | 1.7214 | 4843.6 | 0.0123 |
| | 13 | 1.9606 | 3558.7 | 0.0142 | 1.7325 | 4722.7 | 0.0108 |
| | avg. | 1.8316 | 3267.5 | 0.0102 | 1.6106 | 4631.5 | 0.0061 |
| medium | 1 | 1.9147 | 4177.1 | 0.0139 | 1.5122 | 6674.8 | $1.35 \times 10^{-9}$ |
| | 2 | 1.9255 | 4043.9 | 0.0144 | 1.5694 | 6458.6 | $4.83 \times 10^{-9}$ |
| | 3 | 1.9353 | 3973.19 | 0.0143 | 1.5826 | 6302.2 | $4.91 \times 10^{-10}$ |
| | 4 | 1.9061 | 3683.6 | 0.0130 | 1.6342 | 6027.1 | $5.77 \times 10^{-10}$ |
| | 5 | 1.9308 | 3730.3 | 0.0138 | 1.5920 | 5932.7 | $7.05 \times 10^{-9}$ |
| | 6 | 1.9521 | 3709.0 | 0.0149 | 1.5421 | 5821.9 | $7.02 \times 10^{-10}$ |
| | 7 | 1.9346 | 3644.2 | 0.0140 | 1.6711 | 5977.9 | $2.52 \times 10^{-9}$ |
| | 8 | 1.9398 | 3589.2 | 0.0156 | 1.6162 | 5592.9 | $4.04 \times 10^{-9}$ |
| | 9 | 1.9194 | 3450.6 | 0.0134 | 1.6726 | 5536.0 | $1.65 \times 10^{-9}$ |
| | 10 | 1.9147 | 3400.2 | 0.0138 | 1.6612 | 5442.3 | $1.89 \times 10^{-9}$ |
| | 11 | 1.8947 | 3289.2 | 0.0124 | 1.6824 | 5247.4 | $6.48 \times 10^{-10}$ |
| | 12 | 1.8741 | 3235.7 | 0.0128 | 1.6787 | 5170.6 | $4.94 \times 10^{-9}$ |
| | 13 | 1.9127 | 3273.5 | 0.0136 | 1.6922 | 5078.5 | $2.03 \times 10^{-9}$ |
| | avg. | 1.7394 | 3395.3 | 0.0065 | 1.8162 | 6492.8 | $3.96 \times 10^{-9}$ |
| low | 1 | 1.8671 | 2776.2 | 0.0128 | 1.9440 | 4010.7 | 0.0133 |
| | 2 | 1.8622 | 2747.8 | 0.0123 | 1.9511 | 3916.4 | 0.0143 |
| | 3 | 1.8591 | 2687.6 | 0.0119 | 1.9518 | 3800.5 | 0.0142 |
| | 4 | 1.8464 | 2634.8 | 0.0125 | 1.9443 | 3723.8 | 0.0140 |
| | 5 | 1.8348 | 2589.7 | 0.0116 | 1.9654 | 3711.3 | 0.0143 |
| | 6 | 1.8037 | 2506.4 | 0.0112 | 1.9438 | 3574.6 | 0.0142 |
| | 7 | 1.8096 | 2483.2 | 0.0111 | 1.9327 | 3502.3 | 0.0134 |
| | 8 | 1.8122 | 2476.8 | 0.0111 | 1.9440 | 3484.7 | 0.0138 |
| | 9 | 1.8269 | 2560.7 | 0.0120 | 1.9279 | 3439.4 | 0.0137 |
| | 10 | 1.8182 | 2517.6 | 0.0112 | 1.9385 | 3468.6 | 0.0137 |
| | 11 | 1.8103 | 2487.7 | 0.0114 | 1.9409 | 3447.6 | 0.0141 |
| | 12 | 1.8175 | 2489.8 | 0.0115 | 1.9239 | 3352.0 | 0.0130 |
| | 13 | 1.8047 | 2456.5 | 0.0112 | 1.9238 | 3288.1 | 0.0131 |
| | avg. | 1.7857 | 2506.4 | 0.0107 | 1.8332 | 3449.6 | 0.0100 |

As for the 2D experiments, the residual saturation is relatively small, $S_r < 4\%$, and mostly increases minimally with mineralization. For the fitted Brooks–Corey relations, values for $\lambda$ decrease with mineralization for both the medium-mineralization and high-mineralization columns, contrary to the behavior of the 2D samples and the low-mineralization column, for which $\lambda$ increases with mineralization. However, $p_e$ shows similar behavior to the 2D samples and increases by up to 50%, more for the medium-mineralization column and a bit less for the high-mineralization column. For the Van Genuchten relations, the fitted parameter values behave as expected based on the 2D samples, with the values for $m$ increasing and the values for $\alpha$ and $n$ decreasing with mineralization.

**Table 6.** Parameters of the Van Genuchten capillary pressure–saturation relation (Equation (5)) for the initial and mineralized 3D experiments.

| Column | Section | Initial | | | | Mineralized | | | |
|---|---|---|---|---|---|---|---|---|---|
| | | $\alpha$ [1/Pa] | $m$ [-] | $n$ [-] | $S_r$ [-] | $\alpha$ [1/Pa] | $m$ [-] | $n$ [-] | $S_r$ [-] |
| high | 1 | $3.21 \times 10^{-4}$ | 0.2492 | 7.7823 | 0.0131 | $1.92 \times 10^{-4}$ | 0.3917 | 5.5995 | 0.0355 |
| | 2 | $3.10 \times 10^{-4}$ | 0.2878 | 6.8494 | 0.0140 | $2.00 \times 10^{-4}$ | 0.3978 | 5.3525 | 0.0361 |
| | 3 | $3.02 \times 10^{-4}$ | 0.2699 | 7.2993 | 0.0143 | $1.94 \times 10^{-4}$ | 0.3490 | 5.8283 | 0.0305 |
| | 4 | $2.90 \times 10^{-4}$ | 0.2609 | 7.6448 | 0.0147 | $1.79 \times 10^{-4}$ | 0.4188 | 5.0003 | 0.0290 |
| | 5 | $2.98 \times 10^{-4}$ | 0.2760 | 7.1750 | 0.0141 | $1.84 \times 10^{-4}$ | 0.4106 | 5.0920 | 0.0314 |
| | 6 | $2.89 \times 10^{-4}$ | 0.2902 | 6.8597 | 0.0140 | $1.75 \times 10^{-4}$ | 0.2350 | 7.9835 | 0.0170 |
| | 7 | $2.67 \times 10^{-4}$ | 0.2730 | 7.2652 | 0.0159 | $1.86 \times 10^{-4}$ | 0.2851 | 5.9652 | 0.0133 |
| | 8 | $2.79 \times 10^{-4}$ | 0.2883 | 7.0018 | 0.0144 | $1.68 \times 10^{-4}$ | 0.4181 | 5.0448 | 0.0246 |
| | 9 | $2.69 \times 10^{-4}$ | 0.2871 | 7.0772 | 0.0151 | $1.75 \times 10^{-4}$ | 0.3499 | 5.7250 | 0.0305 |
| | 10 | $2.66 \times 10^{-4}$ | 0.2710 | 7.4726 | 0.0151 | $1.88 \times 10^{-4}$ | 0.3008 | 6.7191 | 0.0296 |
| | 11 | $2.61 \times 10^{-4}$ | 0.3164 | 6.5108 | 0.0152 | $1.76 \times 10^{-4}$ | 0.3729 | 5.4156 | 0.0220 |
| | 12 | $2.65 \times 10^{-4}$ | 0.2973 | 6.8378 | 0.0152 | $1.75 \times 10^{-4}$ | 0.3790 | 5.5501 | 0.0267 |
| | 13 | $2.70 \times 10^{-4}$ | 0.3243 | 6.2505 | 0.0152 | $1.76 \times 10^{-4}$ | 0.4574 | 4.6769 | 0.0245 |
| | avg. | $3.01 \times 10^{-4}$ | 0.2303 | 8.0613 | 0.0106 | $1.92 \times 10^{-4}$ | 0.2942 | 6.2804 | 0.0173 |
| medium | 1 | $2.20 \times 10^{-4}$ | 0.3571 | 5.8343 | 0.0175 | $1.30 \times 10^{-4}$ | 0.2619 | 6.6258 | 0.0065 |
| | 2 | $2.31 \times 10^{-4}$ | 0.3324 | 6.1855 | 0.0170 | $1.24 \times 10^{-4}$ | 0.3978 | 5.2982 | 0.0288 |
| | 3 | $2.36 \times 10^{-4}$ | 0.3178 | 6.4595 | 0.0165 | $1.29 \times 10^{-4}$ | 0.3812 | 5.4375 | 0.0259 |
| | 4 | $2.59 \times 10^{-4}$ | 0.2979 | 6.6594 | 0.0144 | $1.34 \times 10^{-4}$ | 0.4446 | 4.8571 | 0.0242 |
| | 5 | $2.55 \times 10^{-4}$ | 0.3298 | 6.1256 | 0.0153 | $1.30 \times 10^{-4}$ | 0.4877 | 4.6750 | 0.0340 |
| | 6 | $2.57 \times 10^{-4}$ | 0.3232 | 6.2992 | 0.0163 | $1.42 \times 10^{-4}$ | 0.3767 | 5.2589 | 0.0250 |
| | 7 | $2.63 \times 10^{-4}$ | 0.3067 | 6.5422 | 0.0152 | $1.30 \times 10^{-4}$ | 0.5525 | 4.1624 | 0.0250 |
| | 8 | $2.68 \times 10^{-4}$ | 0.2990 | 6.7040 | 0.0167 | $1.45 \times 10^{-4}$ | 0.4906 | 4.2322 | 0.0201 |
| | 9 | $2.82 \times 10^{-4}$ | 0.2678 | 7.3348 | 0.0141 | $1.55 \times 10^{-4}$ | 0.3954 | 5.0102 | 0.0124 |
| | 10 | $2.86 \times 10^{-4}$ | 0.2595 | 7.5419 | 0.0145 | $1.48 \times 10^{-4}$ | 0.4415 | 5.0140 | 0.0244 |
| | 11 | $2.98 \times 10^{-4}$ | 0.2498 | 7.7142 | 0.0129 | $1.53 \times 10^{-4}$ | 0.4774 | 4.6613 | 0.0219 |
| | 12 | $3.03 \times 10^{-4}$ | 0.2535 | 7.5161 | 0.0133 | $1.58 \times 10^{-4}$ | 0.4521 | 4.7626 | 0.0195 |
| | 13 | $3.00 \times 10^{-4}$ | 0.2333 | 8.3231 | 0.0140 | $1.67 \times 10^{-4}$ | 0.3849 | 5.3922 | 0.0157 |
| | avg. | $2.88 \times 10^{-4}$ | 0.2175 | 8.1521 | 0.0073 | $1.28 \times 10^{-4}$ | 0.5193 | 4.0577 | $1.60 \times 10^{-13}$ |
| low | 1 | $3.59 \times 10^{-4}$ | 0.1885 | 9.9331 | 0.0128 | $2.32 \times 10^{-4}$ | 0.3540 | 5.8745 | 0.0158 |
| | 2 | $3.62 \times 10^{-4}$ | 0.1739 | 10.745 | 0.0124 | $2.41 \times 10^{-4}$ | 0.3148 | 6.5318 | 0.0162 |
| | 3 | $3.71 \times 10^{-4}$ | 0.1805 | 10.323 | 0.0119 | $2.49 \times 10^{-4}$ | 0.3391 | 6.0447 | 0.0158 |
| | 4 | $3.78 \times 10^{-4}$ | 0.1925 | 9.6110 | 0.0125 | $2.55 \times 10^{-4}$ | 0.3357 | 6.0589 | 0.0155 |
| | 5 | $3.85 \times 10^{-4}$ | 0.1690 | 10.868 | 0.0116 | $2.57 \times 10^{-4}$ | 0.3173 | 6.4465 | 0.0155 |
| | 6 | $3.98 \times 10^{-4}$ | 0.1665 | 10.8496 | 0.0112 | $2.69 \times 10^{-4}$ | 0.3100 | 6.4836 | 0.0152 |
| | 7 | $4.02 \times 10^{-4}$ | 0.1531 | 11.833 | 0.0111 | $2.76 \times 10^{-4}$ | 0.2988 | 6.6550 | 0.0143 |
| | 8 | $4.03 \times 10^{-4}$ | 0.1801 | 10.072 | 0.0111 | $2.77 \times 10^{-4}$ | 0.3097 | 6.4609 | 0.0146 |
| | 9 | $3.89 \times 10^{-4}$ | 0.1794 | 10.2049 | 0.0120 | $2.81 \times 10^{-4}$ | 0.3101 | 6.3960 | 0.0145 |
| | 10 | $3.96 \times 10^{-4}$ | 0.1682 | 10.8315 | 0.0112 | $2.78 \times 10^{-4}$ | 0.3227 | 6.1956 | 0.0145 |
| | 11 | $4.01 \times 10^{-4}$ | 0.1777 | 10.1999 | 0.0114 | $2.81 \times 10^{-4}$ | 0.2926 | 6.8033 | 0.0149 |
| | 12 | $4.01 \times 10^{-4}$ | 0.1670 | 10.9039 | 0.0116 | $2.91 \times 10^{-4}$ | 0.2557 | 7.6698 | 0.0136 |
| | 13 | $4.06 \times 10^{-4}$ | 0.1826 | 9.9037 | 0.0112 | $2.96 \times 10^{-4}$ | 0.2898 | 6.7747 | 0.0136 |
| | avg. | $3.98 \times 10^{-4}$ | 0.1633 | 10.9521 | 0.0108 | $2.82 \times 10^{-4}$ | 0.2587 | 7.2504 | 0.0108 |

The parameter values for the Brooks–Corey and the Van Genuchten relations of the sections vary from a few to 50% along the sections of the columns, not taking into account the residual saturations, which are at times very small, and, thus, even small absolute value changes result in a large relative variation. Mineralization increases the parameter-value variation between sections, except for the high-mineralization Brooks–Corey parameter $p_e$, for which the variation between the parameter values of the sections decreases slightly. For the Brooks–Corey relations, the column-average $S_r$ and $\lambda$ are each smaller than for the individual sections, with the exception of the medium-mineralization column, which for all sections as well as the average has $S_r < 10^{-8}$, and $\lambda$ is larger than each of the individual sections. The $p_e$ of the column-average Brooks–Corey relations are in the range of the $p_e$ of the individual sections, usually slightly higher than the lowest values. Again, the averaged mineralized medium-mineralization column is the exception with $p_e = 6493\,\text{Pa}$, which is close to the highest values of the sections (6675 Pa) of this column. What can also be

seen in Table 5 is that the Leverett-scaling factors given in Table 4, calculated based on the experimentally measured porosities and permeabilities, greatly overestimate the increase in capillary pressure due to EICP. The mineralized entry pressures of the 3D samples are in no case even close to being twice the value of the unmineralized entry pressure and, thus, the capillary pressure, in general, for the mineralized samples is never as high as predicted by the Leverett-scaling factors from 7.3 to 43.6.

For the Van Genuchten relations, the column-average parameter values are also relatively often the minimal or maximal values when compared to the parameter values of the individual sections of each column. The column-average $S_r$ has the lowest values for all samples except the mineralized high-mineralization column, where two individual sections have slightly lower residual saturations. The column-average $\alpha$ is mostly bound by the $\alpha$ values of the different column sections, except for the mineralized medium-mineralization column, for which the column-average $\alpha$ is slightly lower than the lowest of the sections. The column-average $m$ values are lower than those of the sections for the initial high- and medium-mineralization samples and second-lowest for both the initial and mineralized low-mineralization sample. For the mineralized high–mineralization sample, the column-average $m$ value is the third lowest of the samples' $m$ values, and, for the mineralized medium-mineralization samples, the column-average $m$ value is the second highest. The column-average $n$ values are higher than those of most of the sections, respectively, of both the initial and mineralized samples, but only higher than any section value for the initial high-mineralization sample. An exception is, again, the mineralized medium-mineralization sample, where the $n$ value is actually lower than any of the sections' $n$ values of the sample.

## 4. Discussion

Both the 2D and 3D experiments investigated showed a noticeable increase in capillary pressure for a given saturation due to the precipitation of calcium carbonate during EICP. Qualitatively, both the 2D and the 3D experiments show similar tendencies, but, for the 3D systems, the effects are more difficult to quantify and parameterize. We suspect that this is mainly due to two reasons: the initial unmineralized 3D samples not being as identical as the 2D samples and having only the initial and the final geometries without further temporal resolution. However, the 2D samples are limited to a 2D geometry, which results not only in high initial porosity, much higher than the porosities of most field-relevant porous media but also in an artificially constricted 2D flow field in which even a few pore throats reduced in diameter may already significantly reduce flow or increase the sample's capillary entry pressure. Thus, the relations fitted to the 2D samples are likely not representative of realistic porous media; however, they provide insight into development during mineralization over time, data that were not available for our more realistic 3D samples.

The changes in capillary pressure–saturation relations due to EICP manifest in an increase in the entry pressure $p_e$ of up to a factor of almost 2 for the 3D and 3.6 for the 2D samples. For the 2D samples, the parameter $\lambda$ increased with reducing porosity; for the 3D samples, $\lambda$ increased for two samples, while it decreased for the third sample. Similarly, for a Van Genuchten-type parameterization, the parameter $m$ is increased by EICP for the 2D samples by up to a factor of 10. The parameter $\alpha$ decreased by factors of up to 2.2 for the 3D and 3.1 for the 2D samples, while $n$ decreased by a factor of up to 2.0 for the 3D and 5.8 for the 2D samples. For both parameterizations, the residual saturation $S_r$ changes only slightly. However, for all samples investigated, $S_r$ is small even for highly mineralized geometries. This could be an artifact of the relatively regular, homogeneous initial porous media studied, with the 2D samples having a constant height and the 3D samples consisting of sintered, monodisperse-sized glass beads. The low residual saturations obtained in this study using PNM can be confirmed using direct numerical pore-scale modeling directly on the segmented images to avoid potential artifacts due to the simplification in pore geometry that PNM relies on. Especially for the initial unmineralized samples, the 13 sections seem to

be sufficiently large to be considered an REV, as the parameters determined for the average over the entire 3D sample are relatively similar compared to the parameters determined for the individual sections. Even for mineralized samples, the amount of heterogeneity is still relatively low, at least that which can be captured by the resolutions used. Further, disconnected pore volumes cannot be accounted for in the pore-network model used due to the numerical instabilities resulting from disconnected networks, due to which any disconnected pores needed to be removed. Such disconnection of pores was not observed for the 2D samples and only for a limited amount of pores for the 3D samples. However, for the 3D samples, where some pores became disconnected, the $S_r$ may likely be underestimated, but, to complicate the issue further, disconnection in our study may also be an artifact of the resolution in the case that the remaining pore throats were reduced to diameters smaller than the resolutions of the µXRCT scans, which would result in an assumption of disconnection in our analysis while remaining connected in reality. While the changes in the parameter values of the Van Genuchten relation with reducing porosity have identical trends with reducing porosity for both the 2D and 3D column samples, the differences between, e.g., the high- and medium-mineralization 3D samples make a description of this change based solely on the porosity change impossible. Including the effect of ICP on pore-scale parameters, such as, e.g., pore-size distributions, would likely help explain the effects of ICP on the capillary pressure–saturation curves observed. However, pore-size distributions and other pore-scale parameters are not available on the REV scale, at which real ICP applications will ultimately have to be modeled, adding not only an additional layer of complexity but also the need to upscale the effects of ICP from the pore scale to the REV scale. Including an increased number of REV-scale parameters, such as, e.g., specific surface areas, may potentially improve the predictability of relations for REV-scale capillary pressure–saturation relations of porous media being mineralized using ICP. Then again, such parameters, or their change due to ICP, are themselves difficult to define on the REV scale alone and must be upscaled themselves from the pore scale. Surface areas can directly determine the specific surface areas or similar parameters based on the segmented images.

With our current experimental data, we are reluctant to claim that we determined the final parameterization of the effect of ICP on REV-scale capillary pressure–saturation relations. However, what we can show is that Leverett scaling is not accurate in describing the changes in the capillary pressure–saturation relations due to ICP, especially for the 2D samples, for which ICP completely alters the capillary pressure–saturation relations (see Tables 1 and 2). As seen in Equation (7), Leverett scaling results in a scaling of the capillary pressure–saturation relation by a constant factor, which does not fit the experimental observations of a changing Brooks–Corey parameter $\lambda$ or Van Genuchten parameters $m$ and $n$ (see Tables 5 and 6). To some extent, however, Leverett scaling may still be a simple and good enough parameterization of the effect of ICP on the capillary pressure–saturation relations. In the 3D samples, which provide a more realistic initial geometry, the change in, e.g., the Brooks–Corey relation's parameters $\lambda$ is small, <10%, and, for some samples, $\lambda$ increased, while, for others, $\lambda$ decreased with reducing porosity, creating some uncertainty about how to reliably predict the effect of ICP on a porous medium's $\lambda$. Thus, only the parameter $p_e$ changed significantly, which can be represented using Leverett scaling. However, the changes in capillary pressure and $p_e$, in particular, are much smaller than predicted by Leverett scaling based on experimentally determined porosities and permeabilities. For the high-mineralization sample, Leverett scaling using Equation (7) overestimates the increase in capillary pressure by more than one order of magnitude. To some extent, relations similar to Leverett scaling might still be a simple and good enough parameterization of the effect of ICP on the capillary pressure–saturation relations, as $p_e$ seems to be the parameter most influenced by ICP. The changes in $p_e$ are, however, variable and, for the 3D samples, cannot be described solely based on the porosity reduction, which further complicates the description. Looking at the data of individual

sections of the 3D samples, the variability increases, and the porosity reduction has even less predictive potential.

## 5. Conclusions

Using our workflow, we show that the combination of mineralization experiments with imaging and numerical analysis of the effect of the observed changes in geometry using PNM is able to quantify the resulting changes in the capillary pressure–saturation curves due to ICP. Brooks–Corey and Van Genuchten capillary pressure–saturation relations were fitted to the obtained capillary pressure–saturation curves for both the 2D and 3D samples. However, the change in the parameters of the capillary pressure–saturation relations due to ICP can be parameterized depending on the porosity change only for the 2D samples, while the relations obtained by mineralizing the 3D samples were too variable to be described solely based on porosity change. With the current data, it is difficult to determine whether this variability in the effect of ICP on the capillary pressure–saturations of the 3D samples is a fundamental issue in bridging the scale between the resolved pore geometry and the upscaled, averaged REV. Additional experimental investigations of setups similar to the 3D columns used in this study, but with, e.g., a temporal resolution of the imaging during mineralization and more homogeneous initial pore-space distribution are likely to be beneficial.

Our samples are, by design and for the ease of observation, not representative of, e.g., bulk cap-rock materials, which so far have much higher initial permeability and pore sizes. Smaller pore sizes, however, quickly become challenging as they require an equally reduced resolution for imaging the system to sufficiently resolve the smaller geometries accurately. Investigating samples with properties closer to porous media with more real-world relevance, or, in the next step, even directly using samples made of such porous media will likely increase the real-world application relevance of the results of future studies. However, real-world-relevant porous media, such as, e.g., sand, sand-, or limestone, are less homogeneous in material and pore-structure composition and the resulting hydraulic properties, likely increasing the difficulty in evaluating and parameterizing the effect of ICP on the hydraulic properties compared to ICP in more uniform glass-bead columns or microfluidic cells.

**Author Contributions:** Conceptualization, J.H. and H.S.; methodology, J.H.; software, F.W., M.R. and J.H.; validation, J.H.; formal analysis, M.R. and J.H.; investigation, L.G. and J.H.; resources, J.H. and H.S.; data curation, F.W., M.R. and J.H.; writing—original draft preparation, L.G. and J.H.; writing—review and editing, F.W. and J.H.; visualization, L.G., F.W., M.R. and J.H.; supervision, J.H. and H.S.; project administration, J.H.; funding acquisition, J.H. and H.S.; All authors have read and agreed to the published version of the manuscript.

**Funding:** This research was funded by Deutsche Forschungsgemeinschaft (DFG, German Research Foundation) Project Number 327154368 (Felix Weinhardt, within SFB 1313), DFG Project Number 380443677 (Johannes Hommel), and DFG Project Number 357361983 (Matthias Ruf, Holger Steeb).

**Data Availability Statement:** The image-analysis data, segmentation data, pore-network extraction data, and capillary pressure–saturation relation fitting data are available at https://doi.org/10.18419/darus-2791 (2D experiments, published on 3 August 2022) [60] and https://doi.org/10.18419/darus-1713 (3D experiments, published on 3 August 2022) [59]. The µXRCT image data (projection datasets, reconstructed datasets, meta data, as well as segmented datasets) that support the findings of this study are openly available in the Data Repository of the University of Stuttgart (DaRUS) at: http://doi.org/10.18419/darus-2906, http://doi.org/10.18419/darus-2907, http://doi.org/10.18419/darus-2908 [56–58], published on 12 July 2022.

**Conflicts of Interest:** The authors declare no conflict of interest. The funders had no role in the design of the study; in the collection, analyses, or interpretation of data; in the writing of the manuscript; or in the decision to publish the results.

## Abbreviations

The following abbreviations are used in this manuscript:

| | |
|---|---|
| ICP | Induced Carbonate Precipitation |
| EICP | Enzymatically Induced Carbonate Precipitation |
| PN | Pore Network |
| PNM | Pore-Network Modeling |
| REV | Representative Elementary Volume |
| μXRCT | Micro X-ray Computed Tomography |

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
