# Peer review of "Effects of Enzymatically Induced Carbonate Precipitation on Capillary Pressure–Saturation Relations"

_minerals, doi:10.3390/min12101186_

Round 1

Reviewer 1 Report

Dear authors, 

Very important research and paper analysing the effect of carbonate precipitation in pore systems on the capillary pressure characteristics. We can observe a long, continuous and rigorous work conducted by the authors.

For the present manuscript, I would like to put the following considerations:

i. The residual fluid saturation Sr (irreducible saturation of the wetting fluid) is relatively small even for highly mineralized systems (2D or 3D); In my mind, although the unmineralized system is simple, the layer of mineral precipitate introduce a sort of complexity conducting to trapping mechanisms of wetting fluid. I´am concerning with the simplifications introduced by the pore network model used masking trapping effects. Pore network is extensively used with success for simulating intrinsec permeability, but in this case the roughness/irregularity of the pore surface has not important influence as in capillary pressure (or relative permeability).

In this sense I reccomend to authors cite, in the introduction of the manuscript, the method for simulate primary drainage based on pore morphology. This method is applied directly on the image (2D or 3D) and is not computationaly expensive.

See, for example: 

1)Modelling two-phase equilibrium in three-dimensional porous microstructures, FS Magnani, PC Philippi, ZR Liang, CP Fernandes International journal of multiphase flow 26 (1), 99-123, 2000;

2)Markus Hilpert, Cass T. Miller, Pore-morphology-based simulation of drainage in totally wetting porous media,

Advances in Water Resources, Volume 24, Issues 3–4,

2001,Pages 243-255;

3) Xin Liu, Annan Zhou, Shui-long Shen, Jie Li, Modeling drainage in porous media considering locally variable contact angle based on pore morphology method, Journal of Hydrology,

Volume 612, Part B,2022.

ii) Authors are careful about the not so complex pore structures analysed. I also recommend including comments about size of pores, thinking on cap rocks applications.

Author Response

Dear Reviewer,

we thank you for your valuable comments and suggestions that helped us improve our manuscript. For our detailed answer, we refer to the attached pdf.

Sincerely,
Johannes Hommel (on behalf of the authors)

Reviewer 2 Report

Title: Effects of enzymatically induced carbonate precipitation on capillary pressure-saturation relations

In this manuscript, pore-scale resolved microfluidics experiments in 2D glass cells and 3D sintered glass-bead columns were conducted. Meanwhile, the change in pore geometry was observed. Then, the effect of the geometry change on the capillary pressure-saturation curves were evaluated by numerical drainage experiments using pore-network modeling on pore networks extracted from the observed geometries. Finally, parameters of both Brooks-Corey and Van Genuchten relations were fitted to the capillary pressure-saturation curves determined by pore-network modeling and compared with the reduction in porosity as an average measure of the pore geometry change due to induced precipitation. As the reviewer, my decision is Received, and I have some suggestions and requirements for its improvements.

1.     Line 14: ‘the change in pore geometry observed by light microscopy and X-ray computed tomography’ should be written as ‘the change in pore geometry was observed by light microscopy and X-ray computed tomography’.

2.     In Introduction, it is no need to the introduce the research content detailedly. Meanwhile, the innovation point should be clearly presented in Introduction.

3.     Whether PN modeling should be written as PNM?

4.     The font size of unit should be consistent with text.

5.     For FDK, please give the full name. When a term only appear once in the paper, it is not appropriate to give the abbreviation.

6.     For 2D sample, why two type of experiments with different image resolutions are conducted? And, why this processing method are not adopted in 3D experiment?

7.     The titles of Figure 5 and Figure 6 are the same. Moreover, in the text, some analyses about Figure 5 and Figure 6 should be added to furtherly describe the influence of porosity reduction on capillary pressure-saturation relation.

8.     In the Discussion and Conclusion, the Leverett scaling of capillary pressure-saturation relation is analysed. However, the experimental data has not been fitted through Equation (7) in the text. So, it is better to show the Leverett scaling of capillary pressure-saturation relation through figures, and then compared with Equation (4) and Equation (5)  

9.     It is better to separate the Discussion and Conclusion as two individual parts.

10.   Recent works may be helpful:

* Dynamic capillary pressure analysis of tight sandstone based on digital rock model.

* A comprehensive review of pore scale modeling methodologies for multiphase flow in porous media

Author Response

Dear Reviewer,

thank you for your comments and suggestions helping us to improve our manuscript. For a detailed response we refer to the attached pdf.

Sincerely,
Johannes Hommel (on behalf of the authors)

Reviewer 3 Report

In this study, the effects of enzymatically induced carbonate precipitation on capillary pressure-saturation are explored to illustrate effects of induced precipitation on multi-phase flow parameters.

The study is well organized and flows nicely. The language is beyond publish level. 

Minor revision is recommended and the detailed comments are as below:

(1) the abstract should be shortened. The motivation part is not necessary to state in detail here.

(2) The discussion and conclusion part should be separated. In conclusion part, only clear, short argument points should be stated.

(3) The acronyms, such as XRCT and REV, could be explained when they occur at the first time. I know they are at the end part of the paper, but it could be more convenient in the text for readers.

(4) The unit of the equation should be provided.

Author Response

(The authors gave the same response as above.)

Reviewer 4 Report

The authors experimentally and numerically investigated the effects of enzymatically induced carbonate precipitation on capillary pressure-saturation relations. Generally speaking, this manuscript is well-written and covers a topic of interest to the journal’s readership. These results offer a valuable understanding of subsurface fluid storage. I would like to recommend its publication after a minor revision. Some suggestions below should be carefully considered.

1. In Fig.1, the sample preparation and sample hold are not clear enough. The close-ups are needed for these samples.

2. A photo introducing the detailed experimental set (2D/3D samples) is required to understand this work better.

3. Fig.3 contains different colors of the glass, void, and CaCO3. It would be better if the authors could clarify these elements corresponding to the color in Fig.3.

4. When fluids are transported at the micro/nanoscale porous media, the microscopic effects are significant. The permeability relies not only on porosity but also on the pore pressure. The authors should add some discussion regarding the fluids transport at the micro/nanoscale porous media (doi.org/10.1021/acs.energyfuels.0c03529; doi.org/10.1016/j.fmre.2021.12.010) to references.

Author Response

Dear Reviewer #4,

we thank you for the detailed comments on how to improve the figures. The detailed point-by-point response can be found in the attached responseToReviewer4.pdf.

Sincerely,
Johannes Hommel

Round 2

Reviewer 2 Report

My questions are well addressed. Thanks.

Author Response

We thank the reviewer for the confirmation that we addressed all comments and suggestions from the first round of review.